# Redefining the Fight Against SCLC: Standards, Innovations, and New Horizons

**DOI:** 10.3390/cancers17132256

**Published:** 2025-07-07

**Authors:** Marcel Kemper, Lea Elisabeth Reitnauer, Georg Lenz, Georg Evers, Annalen Bleckmann

**Affiliations:** 1Department of Medicine A, University Hospital Muenster, 48149 Muenster, Germany; leaelisabeth.reitnauer@ukmuenster.de (L.E.R.); georg.lenz@ukmuenster.de (G.L.); georg.evers@ukmuenster.de (G.E.); annalen.bleckmann@ukmuenster.de (A.B.); 2West German Cancer Center, University Hospital Muenster, 48149 Muenster, Germany

**Keywords:** small cell lung cancer, immunotherapy, antibody-drug conjugates, molecular subtypes, targeted therapy, prognosis

## Abstract

Small cell lung cancer is an aggressive malignancy characterized by a poor prognosis. Standard treatment modalities include surgery, radiotherapy, chemotherapy, and immunotherapy; however, therapeutic responses are often limited in duration. Emerging therapeutic strategies involve the use of bispecific antibodies, antibody–drug conjugates, targeted therapies, and personalized treatment approaches guided by molecular biomarkers. Recent advancements in molecular subtyping and novel therapeutics hold promise for ultimately improving patient survival.

## 1. Introduction

### 1.1. Epidemiology

Lung cancer remains the leading cause of cancer-related mortality worldwide, accounting for approximately 1.8 million deaths in 2022 [1], with tobacco smoking as the most significant risk factor. In the United States (U.S.), the proportion of small cell lung cancer (SCLC) among newly diagnosed lung cancer cases declined from 14.5% in 2000 to 11.8% in 2019, with consistent reductions observed across all sex and racial groups, likely due to decreasing smoking rates and improved tobacco control measures [2]. Despite these trends, survival outcomes for SCLC have shown minimal improvement over the past two decades [2,3]. Depending on the disease stage at diagnosis, the 5-year relative survival rate for SCLC ranges from <5% to 25% [4]. Although global smoking rates are declining, some regions, including North Africa, the Middle East, and sub-Saharan Africa, continue to experience a rise in tobacco use [5]. In parallel, increased exposure to occupational carcinogens and air pollution has become a growing concern, particularly in emerging countries with limited healthcare infrastructure [6]. However, the specific contribution of these risk factors to SCLC remains unclear, as available data often do not differentiate between lung cancer subtypes. According to the American Cancer Society, an estimated 226,650 new cases of lung cancer are expected in the U.S. in 2025, with approximately 30,000 attributed to SCLC and a median age at diagnosis of approximately 69 years. At the time of diagnosis, approximately 70% of SCLC cases present with distant metastases, while only about 30% are diagnosed at a limited stage [2,7]. These numbers highlight the ongoing burden of SCLC and the need for better strategies in early detection and treatment.

### 1.2. Pathology

SCLC develops from neuroendocrine cells of the bronchi. Exposure to carcinogens induces mutations in tumor suppressor genes and/or proto-oncogenes. A hallmark of SCLC is the presence of biallelic inactivating mutations in the two tumor suppressor genes Tumor Protein p53 (TP53) and Retinoblastoma 1 (RB1) [8,9]. In addition, mutations in the Neurogenic Locus Notch Homolog (NOTCH) gene family or amplifications of the Myelocytomatosis (MYC) gene are frequently observed in SCLC [8,10].

Microscopically, SCLC typically presents as a small, blue, round cell tumor. The tumor cells are small and have a high nuclear-to-cytoplasmic ratio. The nuclei are hyperchromatic with finely granular chromatin (the so-called “salt and pepper” pattern). Due to the high mitotic activity, apoptosis is frequent, and extensive necrotic areas are commonly observed. To confirm the diagnosis, neuroendocrine markers are typically detected using immunohistochemistry. SCLC tumor cells are usually positive for CD56 (Neural Cell Adhesion Molecule [NCAM]), synaptophysin, chromogranin A, and Thyroid Transcription Factor-1 (TTF-1). A characteristically high Ki-67 proliferation index of over 80% is often observed. Due to the high proliferative activity, blood levels of neuron-specific enolase (NSE) and lactate dehydrogenase (LDH) are often elevated.

### 1.3. Diagnosis and Staging

In its early stages, SCLC often remains asymptomatic [11]. Nonspecific symptoms include fatigue, weight loss, loss of appetite, cough, and shortness of breath. As the disease progresses, patients may experience hemoptysis, chest pain, and symptoms caused by local tumor infiltration (e.g., hoarseness, stridor, dysphagia, superior vena cava syndrome). Symptoms related to metastases, such as bone pain or neurological abnormalities, can also occur [11]. Paraneoplastic syndromes are relatively common in SCLC. While neuroendocrine syndromes (e.g., Cushing’s syndrome, Syndrome of Inappropriate Antidiuretic Hormone Secretion [SIADH]) result from ectopic hormone production, neurological syndromes such as Lambert-Eaton myasthenic syndrome (present in 1–3% of cases) are immune-mediated [12,13,14]. Interestingly, patients with paraneoplastic syndromes have a better prognosis than those without, which might be due to enhanced efficacy of immunotherapy [15].

Diagnosis is based on a combination of medical history, physical examination, laboratory tests (such as NSE and LDH), imaging studies, and a biopsy, typically obtained through bronchoscopy or computed tomography (CT)-guided procedures. For SCLC in stages I–III, where surgery or radiotherapy may be an option, pulmonary function testing (including Forced Expiratory Volume in 1 Second [FEV1], Vital Capacity [VC], and Diffusing Capacity of the Lung for Carbon Monoxide [DLCO]) is also indicated [16]. The guidelines from the National Comprehensive Cancer Network (NCCN) recommend a diagnostic contrast-enhanced CT of the chest, abdomen, and pelvis for further evaluation [17]. If curative treatment is being considered, a fluorodeoxyglucose positron emission tomography (FDG-PET/CT) scan should be performed. As an alternative, bone scintigraphy may be used. Magnetic resonance imaging (MRI) of the brain is recommended for all patients and preferred over CT scans of the brain. Although staging evaluation is optional in patients with extensive stage and dependent on the clinical situation, patients who are not candidates for curative therapy should still receive appropriate staging, including imaging of the brain [17]. The guidelines from the European Society for Medical Oncology (ESMO) recommend a similar diagnostic work-up [16].

For many years, SCLC was classified into Limited Disease (LD-SCLC) and Extensive Disease (ED-SCLC) based on the Veterans Administration Lung Study in 1957. Limited Disease refers to stages where the primary tumor was confined to one hemithorax, with no evidence of tumor spread beyond potentially present ipsilateral pleural effusion or mediastinal lymph node involvement. All other disease scenarios were categorized as Extensive Disease and were associated with a correspondingly poorer prognosis. Today, SCLC is classified according to the TNM or Union for International Cancer Control (UICC) staging system, which allows for more precise categorization, treatment planning, and prognosis assessment [18,19,20]. In this context, staging follows the standard approach used for other tumors, based on the size and local extent of the primary tumor (T), as well as the presence of lymph node involvement (N) or distant metastases (M) (see Table 1).

## 2. Methods

This narrative review was conducted to provide a comprehensive and up-to-date synthesis of current knowledge on SCLC, with a focus on current standards, molecular subtypes, and emerging therapies (see Figure 1). A selective, non-systematic literature search was performed using PubMed to identify relevant peer-reviewed articles published up to June 2025. Search terms included combinations of keywords such as “SCLC”, “therapy”, “biomarker”, “clinical trial”, and “molecular subtyping”. Only articles published in English were considered. To enhance coverage of ongoing research, additional searches were conducted on ClinicalTrials.gov using similar terms to identify registered, ongoing, and completed clinical trials. Studies were selected based on their relevance to the review’s thematic focus, prioritizing original research, recent clinical trials, and high-quality reviews. In cases of conflicting evidence, we considered the methodological rigor, sample size, and recency of studies and aimed to reflect diverging findings in a balanced manner. Reference lists of key publications were screened for additional sources, and relevant abstracts from major oncology conferences were included when peer-reviewed data were not yet available. Furthermore, current national and international guidelines from the United States and Europe were reviewed and integrated. As this is a non-systematic narrative review, no formal protocol (e.g., PRISMA) was applied.

## 3. Molecular Subtyping

Based on the expression of specific transcription factors, SCLC can be classified into molecular subtypes. This molecular classification enhances our understanding of tumor heterogeneity and may help explain differences in prognosis and therapeutic responses among patients. The four major subtypes are as follows [21]:

-SCLC-A (ASCL1-dominant): Characterized by high expression of Achaete-Scute Family BHLH Transcription Factor 1 (ASCL1), this subtype is associated with classic neuroendocrine features and may be sensitive to B-Cell Lymphoma 2 (BCL-2) inhibition due to its dependency on anti-apoptotic pathways.-SCLC-N (NEUROD1-dominant): Defined by predominant expression of Neurogenic Differentiation Factor 1 (NEUROD1), this subtype exhibits features of neuronal differentiation and often harbors MYC gene amplifications, suggesting potential vulnerability to aurora kinase inhibitors and other MYC-targeted therapies.-SCLC-P (POU2F3-dominant): Marked by high expression of POU Class 2 Homeobox 3 (POU2F3), this non-neuroendocrine subtype represents a distinct tuft-cell-like lineage and may be sensitive to Insulin-like Growth Factor 1 Receptor (IGF1R) inhibitors or other novel targeted agents.-SCLC-I (Inflamed subtype): Distinguished by low expression of traditional lineage-defining transcription factors but high levels of immune cell infiltration and the upregulation of immune response-related genes, SCLC-I tumors appear particularly sensitive to immunotherapy.

SCLC-Y refers to a proposed additional subtype of SCLC characterized by high expression of Yes-Associated Protein 1 (YAP1), a transcriptional co-activator involved in the Hippo signaling pathway. This subtype exhibits less neuroendocrine differentiation than SCLC-N or SCLC-P and is potentially associated with chemotherapy resistance and increased responsiveness to immune checkpoint blockade [22,23]. SCLC-Y also displays more mesenchymal and proliferative features and may possess distinct biological vulnerabilities compared to the other four established SCLC subtypes. However, SCLC-Y remains less well-defined and less widely accepted in the field [24,25]. Some experts suggest that SCLC-Y overlaps with or could be considered a subset of SCLC-I, given its non-neuroendocrine characteristics and stem-like traits.

These molecular subtypes differ not only in gene expression profiles but also in clinical behavior and therapeutic vulnerabilities. Retrospective analyses from the IMpower133 trial and preclinical studies suggest that the SCLC-I subtype may be more responsive to immunotherapy [21,26]. However, conflicting evidence also exists, with some studies reporting no clear association between the tumor immune microenvironment and molecular subtypes [27]. As such, these classifications remain under development and currently lack standardized, clinically validated assays, limiting their immediate clinical utility. Additionally, growing evidence highlights substantial intratumoral heterogeneity [28] and dynamic plasticity between subtypes [29], further challenging the stability and reproducibility of this framework. While molecular subtyping holds promise, prospective validation and translational research are needed to establish its role in clinical decision-making.

## 4. Treatment for Limited Disease (LD-SCLC)

### 4.1. Current Treatment Standards in LD-SCLC

A curative treatment approach is formally possible only for patients with LD-SCLC. The median overall survival (OS) in this stage (with treatment) is approximately 12–18 months, with a 2-year survival rate of about 41% [30]. For patients with LD-SCLC suitable for surgical resection (stage I-II, cT1-2 N0), the guidelines of the NCCN and ESMO recommend primary surgical resection followed by either four cycles of adjuvant platinum-based chemotherapy (in case of R0 resection) or concurrent chemoradiotherapy (cCRT) (in case of R1–2 resection). For patients not suitable for surgical resection (stages I–III, cT1-4 N0-3 M0), subsequent therapy depends on patient performance status (PS). While patients with good PS (0–1) are recommended to receive cCRT, patients with poor PS (≥2) can alternatively receive sequential CRT [16,17,31]. Studies have shown no clear significant difference between conventional fractionation (total dose of 60–66 Gy) and hyperfractionated radiotherapy (total dose of 45 Gy) [32,33,34]. After the completion of cCRT, prophylactic cranial irradiation (PCI) with a total dose of 25–30 Gy should be considered [30]. In LD-SCLC, PCI has prognostic relevance, as it can reduce the risk of central nervous system (CNS) metastases from 40% (without PCI) to 10% (with PCI), although the improvement in 5-year survival is only around 5% [35]. Patients who do not show disease progression following cCRT should also receive consolidation therapy with durvalumab according to the results from the phase III ADRIATIC trial (see Table 2). Here, durvalumab significantly improved both progression-free survival (PFS) and OS compared to placebo. The median OS was 55.9 months for durvalumab. The median PFS was 16.6 months vs. 9.2 months, respectively (Hazard Ratio [HR] = 0.76; 95% Confidence Interval [CI]: 0.61–0.95). Pneumonitis occurred more frequently with durvalumab (38% vs. 30%), though the overall safety profile remained acceptable. Based on these results, durvalumab has been approved as consolidation therapy following cCRT by both the U.S. Food and Drug Administration (FDA) and the European Medicines Agency (EMA). The results in the durvalumab–tremelimumab group remain blinded [36].

### 4.2. Novel Treatment Approaches in LD-SCLC

As only about 30% of SCLC cases are diagnosed at a limited disease stage, most research efforts have focused on optimizing treatment for extensive-stage disease and recurrence. Nevertheless, several agents under investigation in ED-SCLC are also being evaluated in LD-SCLC (see Figure 2 and Table 3). Drawing parallels to non-small cell lung cancer (NSCLC), neoadjuvant chemoimmunotherapy followed by surgery is currently being explored for LD-SCLC. Retrospective analyses suggest promising feasibility and efficacy, though no randomized phase III data are available to date. Ongoing phase II trials (NCT06911606, NCT04539977, NCT04542369) are evaluating this approach, but it cannot yet be considered an alternative treatment option. Whether pathological remission status can serve as a surrogate prognostic biomarker in SCLC, as it does in NSCLC, remains to be determined.

Another phase II study (NCT06719700) is investigating the combination of cCRT with immunotherapy and anti-angiogenic targeted therapy. In parallel, the phase III KEYLYNK-013 trial (NCT04624204) is assessing the addition of olaparib to immunotherapy as maintenance after cCRT. However, concerns over cumulative toxicity may limit the clinical applicability of these approaches, making it unlikely that they will redefine the current standard of care (SOC).

Although results from the durvalumab–tremelimumab arm of the ADRIATIC trial are still pending, targeting both Programmed Death (Ligand) 1 [PD-(L)1] and Cytotoxic T-Lymphocyte Antigen 4 (CTLA-4), has already been investigated in LD-SCLC. The STIMULI trial assessed consolidation therapy with nivolumab and ipilimumab following cCRT but failed to meet its primary endpoint of PFS, largely due to treatment discontinuations caused by immune-related adverse events (irAEs) [42]. Given these toxicity challenges, dual checkpoint inhibition in LD-SCLC appears limited, and attention has shifted towards more tolerable agents such as tarlatamab. Following the promising results from the DeLLphi-304 trial (see Section 5.2.3), tarlatamab is now also being evaluated in LD-SCLC. Therefore, the phase III DeLLphi-306 trial (NCT06117774) is currently investigating tarlatamab versus placebo as consolidation therapy following cCRT.

**Table 3 cancers-17-02256-t003:** Tabular overview of recent clinical trials on novel therapeutic approaches in small cell lung cancer (SCLC). This table summarizes key study data on the investigated therapeutics, including study design, patient cohorts, interventions/dosing, primary efficacy outcomes, safety profiles, and relevant references. Abbreviations used in this table include ADC (antibody-drug conjugate), AE (adverse events), BiTE (bispecific T-cell engager), cCRT (concurrent chemoradiotherapy), CRPC (castration-resistant prostate cancer), CRS (cytokine release syndrome), CTFI (chemotherapy-free interval), CTX (chemotherapy), DLT (dose-limiting toxicities), DCR (disease control rate), DOR (duration of response), ED (Extensive Disease), FL (follicular lymphoma), ICANS (Immune Effector Cell-Associated Neurotoxicity Syndrome), ILD (interstitial lung disease), LD (Limited Disease), mo (months), NEC (neuroendocrine carcinoma), ORR (objective response rate), OS (overall survival), PFS (progression-free survival), pts (patients), SD (stable disease), SOC (standard of care), TEAE (treatment-emergent adverse events), TRAE (treatment-related adverse events), and VOD (veno-occlusive disease).

Agent	Target/ Mechanism	Trial (Phase)	N	Population	Key Results	Safety Profile	Limitations	Clinical Implications	Reference
Tarlatamab	DLL3/BiTE	DeLLphi-301 (Phase II)	220	≥2 L ED-SCLC	ORR: 40%; PFS 4.9 mo; OS: 14.3 mo; 58% DOR ≥ 6 mo	CRS (51%), decreased appetite (29%), pyrexia (35%), ~3% treatment discontinuation	No control group; CRS/ICANS require careful monitoring; hospitalization	Favorable survival and tolerability, FDA approval	[43]
		DeLLphi-304 (Phase III)	~700	Relapsed ED-SCLC after platinum-based CTX	Improvement in OS compared to local SOC (13.6 vs. 8.3 mo)	Lower grade ≥ 3 AEs (54% vs. 80%), fewer discontinuations	Indirect comparison to mix of chemo agents; CRS/ICANS monitoring not fully standardized	New SOC for second-line treatment in ED-SCLC	[44]
		DeLLphi-305 (Phase III)	550	Maintenance after 1 L induction in ED-SCLC without tumor progression	recruiting				
		DeLLphi-306 (Phase III)	400	Consolidation after cCRT in LD-SCLC	recruiting				
Rovalpituzumab Tesirine (Rova-T)	DLL3/ADC	NCT01901653 (Phase I/II)	82	≥1 L SCLC	ORR 18% (38% in pts with high DLL3 expression)	Grade ≥ 3 TRAEs: thrombocytopenia (11%), pleural effusions (8%), elevated lipase (7%)	No control, small sample size; modest efficacy; high-grade toxicities	DLL3 expression linked to response, foundational for future ADCs	[45]
		TAHOE (Phase III)	444	≥1 L Advanced/ metastatic SCLC with high DLL3 expression	Inferior OS compared to topotecan (6.3 vs. 8.6 mo) → trial discontinuation	serosal effusions, photosensitivity, peripheral edema	No efficacy benefit; worse outcome vs. SOC; toxicity of ADC payload	First anti-DLL3 ADC tested in SCLC phase III; discontinuation of Rova-T program	[46]
		MERU (Phase III)	748	Maintenance after 1 L platinum-based CTX in ED-SCLC	DLL3-high tumors: Significant improvement in PFS (4.0 vs. 1.4 mo) but not in OS (8.5 vs. 9.8 mo) → trial termination	≥20% AEs in the Rova-T arm (pleural effusion, decreased appetite, peripheral edema, photosensitivity, fatigue, nausea, dyspnea) More grade ≥ 3 toxicities	No active comparator; high-grade AEs offset PFS gains	First phase III DLL3-targeted ADC in maintenance; discontinuation of Rova-T program	[47]
ZL-1310		NCT06179069 (Phase I)	112	≥1 L ED-SCLC; ±Atezolizumab ±Carboplatin	ORR 68%; pts with brain metastases: ORR 80%, DCR 100%	39% grade ≥ 3 TRAEs (anemia, neutropenia, thrombocytopenia), one DLT	Early phase, small sample size; long-term safety and higher-dose toxicity unresolved	Highly promising early activity, rapid and intracranial responses; manageable early safety	[48]
Sacituzumab Govitecan	Trop2/ADC	NCT03964727/ TROPiCS-03 (Phase II)	227 (43 SCLC)	Recurrent ED-SCLC after 1 L platinum-based CTX and PD(L)1 directed therapy	ORR: 41.9% DOR: 4.7 mo PFS: 4.4 mo OS: 13.6 mo	74.4% grade ≥ 3 TEAEs; no TEAE led to treatment discontinuation; 1 TEAE (neutropenic sepsis) led to death	Single-arm, small sample size; grade ≥ 3 events common; optimal patient selection unclear	Promising second-line option in ED-SCLC	[49]
Ifinatamab Deruxtecan (DS-7300)	B7-H3/ADC	NCT04145622 (Phase I/II);	250 (22 SCLC)	Advanced/unresectable solid tumor (incl. SCLC), that is refractory to or intolerable with standard treatment, or for which no standard treatment is available.	ORR: 52.4%; PFS: 5.6 mo; OS: 12.2 mo	36.4% grade ≥ 3 TEAEs (nausea, decreased appetite, constipation); 22.7% treatment discontinuation	Single-arm; small sample size; ILD/pneumonitis risk unclear; Dose/regimen optimization pending	First-in-human for anti-B7-H3 ADC in SCLC; broad efficacy across SCLC patients	[50]
		NCT06203210/ IDeate-Lung02 (Phase III)	540	Relapsed SCLC after 1 L platinum-based CTX	recruiting				
ABBV-011	SEZ6/ADC	NCT03639194 (Phase I)	99	Relapsed/refractory ED-SCLC after 1 L platinum-based CTX	At 1.0 mg/kg: ORR: 25%; DOR: 4.2 mo; PFS: 3.5 mo	Fatigue (50%), nausea (42%), thrombocytopenia (41%), increased ASAT (22%), 2 cases of VOD	No comparator; modest response in selected cohort; grade ≥ 3 AEs in ~48%; biomarker cutoff and expression variability	First SEZ6-targeted ADC tested in SCLC; No further trials yet initiated	[51]
Olaparib	PARP inhibitor	NCT04728230, PRIO (Phase I/II)	63	1 L ED-SCLC; +durvalumab with carboplatin/etoposide and/or radiotherapy	recruiting				
		NCT04624204 KEYLYNK-013 (Phase III)	672	1 L LD-SCLC; +pembrolizumab post cCRT	Active, not recruiting				
Talazoparib		SWOG S1929 (Phase II)	106	Maintenance after 1 L in SLFN11-positive ED-SCLC; ±atezolizumab	Improvement in PFS (2.9 mo vs. 2.4 mo), but no difference in OS	17% grade ≥ 3 non-hematologic TRAEs, higher rate of hematologic TRAEs (50%)	Phase II, small sample; No OS improvement; Hematologic toxicity in 50%; SLFN11 not validated as predictive for OS	Biomarker-selected, first of its kind in SCLC; requires larger studies to confirm clinical value	[52]
Niraparib		NCT04701307 (Phase II)	48	≥1 L SCLC or NECs; +dostarlimab	Active, not recruiting				
Navitoclax	BCL2 inhibitor	NCT00445198 (Phase II)	39	Recurrent/progressive SCLC after ≥1 L	Early efficacy signs (PR 2.6%, SD 23%) PFS 1.5 mo; OS 3.2 mo	Dose-limiting thrombocytopenia (41% grades 3–4)	Small, early-phase, no comparator; minimal reported efficacy; DLTs led to trial termination	Identified thrombocytopenia as key toxicity	[53]
Venetoclax		NCT0442221, NCT04543916 (Phase Ib/II)	N/A	1 L ED-SCLC, relapsed/refractory SCLC	terminated/withdrawn				
Tazemetostat	EZH2 inhibitor	NCT05353439 (Phase I)	60	Relapsed/recurrent SCLC after platinum-based CTX; +topotecan/pembrolizumab	recruiting				
Mevrometostat		NCT03460977 (Phase I/II)	343	Relapsed/refractory SCLC, CRPC, and FL; ±SOC treatment	So far, no formal efficacy results have been reported for SCLC cohort	SCLC-specific safety outcomes unreported	Early-phase, small cohorts; unreported efficacy and safety data	First EZH1/2 inhibitor tested in SCLC	[54]
Valemetostat	EZH1/2 inhibitor	NCT03879798 (Phase I/II)	22	Recurrent SCLC after platinum-based CTX	ORR: 21% (4/19); DOR: 4.6 mo PFS: 2.2 mo OS: 6.6 mo	≥20% TRAEs were diarrhea, fatigue, nausea, and rash; 3 DLTs → early trial termination	Single-arm, small cohort; modest outcomes; DLTs occurred	Explored SLFN11, epigenetic markers, subtype shifts	[55]
Lurbinectedin	DNA damage	NCT02611024 (Phase I/II)	320 (100 SCLC)	Relapsed/refractory solid tumors, including SCLC; +irinotecan	ORR: 52.7% DOR: 7.6 mo PFS: 5.0 mo OS: 12.7 mo (in pts with CTFI > 20 days)	71.6% grade ≥ 3 TRAEs (neutropenia, anemia, diarrhea, fatigue); 31.1% serious AEs, 6.8% treatment discontinuations	Single-arm, open-label; high grade ≥ 3 AE rate	Proof-of-concept for the combination of lurbinectedin + irinotecan; led to LAGOON phase III trial	[56]
		NCT05153239/ LAGOON (Phase III)	705	Relapsed SCLC after platinum-based CTX with CTFI ≥ 30 days; +irinotecan	Active, not recruiting				
		NCT05091567 IMforte (Phase III)	660	Maintenance after 1 L induction treatment in ED-SCLC without tumor progression	PFS: 5.4 mo OS: 13.2 mo	25.6% grade 3/4 TRAEs; AEs led to treatment discontinuation in 6.2%	Open-label; moderate absolute improvements; PFS still limited; increased serious AEs and some fatal events	Significant survival benefits; potential new option for maintenance therapy	[57]

Despite ongoing trials exploring novel strategies such as neoadjuvant chemoimmunotherapy and consolidation with tarlatamab, their clinical value in LD-SCLC remains uncertain due to limited data and concerns regarding feasibility and toxicity.

## 5. Treatment for Extensive Disease (ED-SCLC)

### 5.1. Current Treatment Standards in ED-SCLC

In advanced-stage SCLC, the standard therapy according to NCCN and ESMO is platinum-based chemotherapy (carboplatin or cisplatin plus etoposide) combined with a PD-L1 inhibitor (atezolizumab or durvalumab) [16,17]. This combination has demonstrated a significant survival benefit in randomized phase III trials (IMpower133 and CASPIAN) and is now considered the international SOC [37,38]. The median OS under chemotherapy is just under 10 months, with a 2-year survival rate of approximately 9% [30]. The addition of immunotherapy has been shown to prolong OS by only about 2 months, from 10 to 12 months [58]. After successful first-line treatment, maintenance therapy with the checkpoint inhibitor is recommended until disease progression. PCI is only considered for selected ED-SCLC patients with a high risk of brain metastases [16]. Since February 2025, a new approval has been granted for the combination of the PD-1-targeting monoclonal antibody serplulimab with cis-/carboplatin and etoposide. This combination demonstrated a significant improvement in OS compared to chemotherapy alone in the randomized phase III ASTRUM-005 trial [39] (see Table 2).

Topotecan is the recommended second-line treatment according to ESMO guidelines [16], though it shows limited efficacy. Alternatively, lurbinectedin can be used [17], based on data from a single-arm phase II basket trial (NCT02454972) published in 2020 that showed promising efficacy [40]. However, lurbinectedin has only been approved by the FDA, while approval by the EMA is still pending.

### 5.2. Novel Treatment Approaches in ED-SCLC

Despite modest improvements with the addition of immunotherapy to chemotherapy, the overall prognosis of SCLC remains poor, particularly in cases of recurrence or disease progression. This underscores the urgent need for novel therapeutic strategies. The following section provides a concise overview of current research directions and recent clinical trial results in this field (see Figure 2 and Table 3).

#### 5.2.1. Lurbinectedin

Lurbinectedin is a synthetic alkaloid analogue with cytotoxic activity that functions as a DNA-binding inhibitor of transcription [59,60]. In relapsed SCLC, lurbinectedin showed activity and had an acceptable safety profile [40]. In addition to its direct cytotoxic effects, lurbinectedin exhibits immunomodulatory properties that further enhance its antitumor activity [61]. Very recently, the results of the phase III, randomized, open-label IMforte trial have been released, which evaluated the efficacy of maintenance therapy with lurbinectedin plus atezolizumab in patients with ED-SCLC who had not progressed after first-line chemoimmunotherapy. The trial demonstrated a significant improvement in PFS, with a median PFS of 5.4 months in the combination arm versus 2.1 months in the atezolizumab monotherapy arm (HR 0.54; *p* < 0.0001). The median OS also improved from 10.6 months to 13.2 months (HR 0.73; *p* = 0.0174), respectively. Although treatment-related adverse events (TRAEs) were more common in the combination arm (83.5% vs. 40%), the regimen was overall well tolerated, with a low rate of treatment discontinuation. These findings suggest that lurbinectedin combined with atezolizumab may represent a promising new maintenance strategy for patients with ED-SCLC [57]. While lurbinectedin is approved by the FDA but not yet by the EMA, for second-line treatment of SCLC, the phase III ATLANTIS trial failed to show an OS benefit for the combination of lurbinectedin and doxorubicin compared to standard regimens [41]. Building on phase II results [56], an ongoing phase III trial (LAGOON) is currently evaluating lurbinectedin with or without irinotecan as second-line treatment for SCLC (NCT05153239). The first results are expected in the first quarter of 2026. Overall, lurbinectedin appears promising for use in maintenance and second-line settings; however, its long-term applicability may be restricted by toxicity, particularly given the high burden of comorbidities in SCLC patients.

#### 5.2.2. Dual Checkpoint Inhibition in SCLC

In the phase I/II CheckMate 032 trial, combining nivolumab (anti-PD-1) with ipilimumab (anti-CTLA-4) led to a modest increase in objective response rate (ORR) compared to nivolumab alone (21% vs. 10%) in patients with recurrent SCLC, albeit with higher toxicity [62]. In the CASPIAN trial, the addition of tremelimumab (anti-CTLA-4) to durvalumab and chemotherapy did not result in a significant survival advantage and led to a higher incidence of serious adverse events (AEs) [63]. These findings suggest that while dual checkpoint blockade may potentiate immune activity against SCLC, its clinical benefit remains modest and is counterbalanced by substantial toxicity. Therefore, these regimens may be appropriate only for a small subset of patients with minimal comorbidities.

#### 5.2.3. Tarlatamab

Tarlatamab is a bispecific T-cell engager (BiTE) that simultaneously binds Delta-like protein 3 (DLL3) on tumor cells and CD3 on T-cells. DLL3 is overexpressed in >80% of SCLC and minimally present in normal tissues, making it an attractive therapeutic target. In the phase II DeLLphi-301 study, tarlatamab (10 mg every two weeks) was administered to patients with advanced, previously treated SCLC and showed notable activity, achieving an ORR of 40%, with 58% of responders maintaining their response for at least six months. AEs such as cytokine release syndrome (CRS), fatigue, and fever were reported, with severe events leading to treatment discontinuation in approximately 3% [43]. The FDA approved tarlatamab in May 2024 for adults with ED-SCLC after progression on platinum-based chemotherapy. These findings were confirmed in the phase III DeLLphi-304 trial, where tarlatamab significantly improved OS compared to chemotherapy (13.6 vs. 8.3 months; HR 0.60; 95% CI, 0.47–0.77; *p* < 0.001). Tarlatamab was also associated with fewer grade ≥ 3 AEs (54% vs. 80%) and lower rates of treatment discontinuations (5% vs. 12%) [44]. An ongoing phase III trial is also evaluating tarlatamab as maintenance therapy after first-line treatment (DeLLphi-305). In summary, tarlatamab is expected to establish itself as the new SOC for second-line SCLC treatment, with EMA approval eagerly anticipated.

#### 5.2.4. Rovalpituzumab Tesirine (Rova-T)

Rova-T is an anti-DLL3 ADC. Early trials showed promising results, with a phase I study reporting an ORR of 18%, and 38% in the subgroup of patients with high DLL3 expression [45]. However, subsequent larger trials were disappointing. The phase III TAHOE study comparing Rova-T to topotecan in second-line therapy was terminated early due to inferior survival in the Rova-T group [46]. The MERU trial demonstrated that Rova-T provided no benefit as a maintenance therapy [47]. Therefore, AbbVie discontinued Rova-T development in August 2019.

#### 5.2.5. ZL-1310

ZL-1310 is a next-generation anti-DLL3 ADC linked to a novel camptothecin derivative (a topoisomerase I inhibitor). In an ongoing phase Ia/b trial (NCT06179069), preliminary results from the dose-escalation phase showed promising activity in previously treated ED-SCLC patients. Objective responses were observed in 19 of 28 patients (68%). Patients with baseline brain metastases had an 80% response rate and 100% disease control rate (DCR) [48]. ZL-1310 was generally well tolerated, with grade ≥ 3 TRAEs reported in 39% of patients, most commonly anemia (21%), neutropenia (18%), thrombocytopenia (11%), and one case of dose-limiting toxicity (DLT). An ongoing phase I trial (NCT06179069) is investigating ZL-1310 as monotherapy and in combination with atezolizumab, with or without carboplatin, for the treatment of ED-SCLC. The agent has received orphan drug designation from the FDA, highlighting its early potential; however, available data remain preliminary and require further validation.

#### 5.2.6. Sacituzumab Govitecan (SG)

This anti-Trop-2 (Tumor-associated calcium signal transducer 2) ADC was evaluated in the phase II TROPiCS-03 trial in patients with recurrent ED-SCLC, administered at 10 mg/kg on days 1 and 8 of a 21-day cycle. ORR was 41.9%, with a median duration of response (DOR) of 4.7 months. The safety profile was manageable. Based on these results, the FDA granted Breakthrough Therapy Designation in December 2024 [49]. The phase III EVOKE-SCLC-04 trial (NCT06801834) is currently evaluating SG as a second-line treatment compared to topotecan; however, enrollment only began in April 2025. It remains unclear how these results will compare once tarlatamab becomes the new SOC in the second-line setting. Additionally, another phase II trial (NCT06667167) is investigating SG as a maintenance therapy in combination with pembrolizumab following first-line induction therapy. In summary, the role of SG in the treatment landscape will need to be further defined.

#### 5.2.7. Ifinatamab Deruxtecan (DS-7300)

This novel ADC targets B7-H3 (B7 Homolog 3, CD276), a transmembrane protein associated with poor prognosis and overexpressed in SCLC. Preclinical data showed strong antitumor activity. In a phase I/II study (NCT04145622), DS-7300 achieved a 52.4% ORR in evaluable SCLC patients and was well tolerated. A phase III trial (IDeate-Lung02, NCT06203210) is now underway to further evaluate its efficacy in relapsed SCLC [50,64].

#### 5.2.8. ABBV-011

ABBV-011 targets Seizure-Related 6 Homolog (SEZ6), a protein overexpressed in SCLC. This ADC combines a SEZ6-specific monoclonal antibody with calicheamicin, linked via a stable, non-cleavable linker. Preclinical studies demonstrated potent activity. In a phase I trial (NCT03639194), ABBV-011 was well tolerated with dose-dependent side effects (fatigue, nausea, thrombocytopenia). At 1.0 mg/kg, the ORR was 25%, with a median DOR of 4.2 months and PFS of 3.5 months. These results support further investigation of SEZ6 as a therapeutic target [51,65].

#### 5.2.9. PARP Inhibitors

Poly ADP-ribose polymerase (PARP) inhibitors are being actively explored in clinical trials for their potential to enhance therapeutic outcomes in SCLC. Building upon preclinical evidence suggesting that PARP inhibition can sensitize SCLC cells to DNA-damaging agents, several studies are assessing combinations of PARP inhibitors with chemotherapy, immunotherapy, and radiation [66]. For example, the ongoing phase I/II PRIO trial (NCT04728230) is evaluating the combination of olaparib with durvalumab, carboplatin, etoposide, and/or radiotherapy in patients with ED-SCLC. Additionally, the SWOG S1929 trial showed that maintenance therapy with talazoparib and atezolizumab improved PFS in patients with Schlafen 11 (SLFN11)-expressing tumors, highlighting the importance of biomarker-driven approaches [52]. Another phase II trial (NCT04701307) is evaluating the combination of niraparib and dostarlimab. Furthermore, Allarity Therapeutics has initiated a phase II trial examining the efficacy of combining stenoparib, a dual PARP and tankyrase inhibitor, with temozolomide in patients with recurrent SCLC, aiming to improve tolerability and therapeutic synergy. However, a key limitation of these approaches is the lack of robust, prospective validation for predictive biomarkers like SLFN11, as well as the need to better define patient subgroups that will truly benefit from PARP inhibitor–based combinations, given the heterogeneous responses and potential for added hematologic and gastrointestinal toxicity in heavily pretreated populations.

#### 5.2.10. BCL2 Inhibitors

BCL-2 inhibitors, including navitoclax and venetoclax, are currently under investigation in SCLC due to the frequent overexpression of BCL-2 in these tumors. Navitoclax (ABT-263) demonstrated limited efficacy as a single agent in a phase II study involving patients with relapsed SCLC, with a partial response (PR) observed in 2.6% of patients and stable disease (SD) in 23%. The median PFS was 1.5 months, and the median OS was 3.2 months. Thrombocytopenia was the most common toxicity, reaching grades 3–4 in 41% of patients [53]. Preclinical studies have shown that venetoclax effectively induces apoptosis in SCLC cells with high BCL-2 expression, indicating potential therapeutic benefit [67]. A phase Ib/II clinical trial aimed to evaluate the safety and efficacy of venetoclax in combination with chemoimmunotherapy in patients with untreated ED-SCLC (NCT04422210), whereas another phase I/II trial aimed to establish the recommended phase II dose for venetoclax when given in combination with irinotecan in patients with relapsed or refractory SCLC (NCT04543916). A major limitation is the limited clinical efficacy and high toxicity observed, along with early trial terminations, which have hindered further evaluation despite promising preclinical data.

#### 5.2.11. Epigenetic Modulators (EZH2 Inhibitors)

In SCLC, EZH2 (Enhancer of Zeste Homolog 2), a histone methyltransferase, is commonly overexpressed and plays a key role in driving tumor progression and resistance to chemotherapy. Ongoing clinical trials are investigating EZH2 inhibitors as potential therapeutic agents in SCLC. A phase I trial (NCT05353439) is investigating whether adding tazemetostat to topotecan and pembrolizumab can improve efficacy in recurrent SCLC. Another investigational agent, mevrometostat (PF-06821497), is being assessed in a phase I trial (NCT03460977) involving patients with SCLC, castration-resistant prostate cancer (CRPC), and follicular lymphoma (FL). The preliminary results indicate that while some patients with FL achieved a PR, SCLC patients experienced disease progression, highlighting the need for further research to optimize treatment strategies [54]. Additionally, valemetostat (DS-3201b) is being evaluated in a phase I/II study for its potential to overcome chemoresistance in SCLC by targeting the EZH2–SLFN11 pathway. Although the combination of valemetostat with irinotecan demonstrated clinical activity, the regimen was not well tolerated due to overlapping toxicities, limiting its further development [55]. Preclinical studies have also explored the use of proteolysis-targeting chimeras (PROTACs) to degrade EZH2, demonstrating the potential to overcome chemo-resistance [68,69]. Collectively, these ongoing trials highlight the potential of EZH2 inhibitors as components of combination therapies to improve outcomes in SCLC. However, in the absence of phase III data, these approaches remain experimental and are currently limited by toxicity and modest efficacy.

## 6. Recommendations and Future Perspectives

Despite recent advances, SCLC continues to present substantial clinical and translational challenges:A major obstacle remains the lack of durable treatment responses, particularly in the relapsed or refractory setting, where most available therapies offer only modest survival benefits. Rapid disease progression and the early development of treatment resistance further complicate clinical management.The absence of reliable predictive biomarkers significantly limits effective patient selection for emerging therapies. While molecular subtyping shows promise, its clinical implementation is hindered by inconsistent classification systems and the lack of standardized assays.Treatment-related toxicity remains a significant barrier, especially in combination regimens involving chemotherapy, immunotherapy, or targeted agents. Such regimens are often poorly tolerated by patients with limited performance status or significant comorbidities.Clinical trial development is also hampered by slow patient accrual, insufficient biomarker stratification, and early treatment discontinuations, collectively contributing to the slow pace of progress in this aggressive malignancy.

Nevertheless, recent therapeutic developments have reinvigorated the treatment landscape, with novel immunotherapies and ADCs showing particular promise:The bispecific T-cell engager tarlatamab has emerged as a leading candidate, demonstrating improved survival and a favorable safety profile in a recent phase III trial. It is anticipated to become a new standard of care in the second-line setting, with potential use in earlier stages, including consolidation therapy for LD-SCLC, currently under investigation.The addition of lurbinectedin to immunotherapy as maintenance therapy in ED-SCLC remains under evaluation; however, its potential to become a new standard is uncertain given concerns that toxicity may outweigh any survival benefit.ADCs such as sacituzumab govitecan and ZL-1310 have shown promising activity in heavily pretreated populations, including patients with brain metastases. Newer-generation ADCs may offer effective alternatives for patients ineligible for more intensive treatment.By contrast, the future of targeted therapies such as PARP-, BCL-2-, and EZH2-inhibitors remains uncertain, due to modest clinical activity, toxicity concerns, and the early termination of several trials.Future efforts should prioritize the validation of predictive biomarkers (e.g., SLFN11, DLL3, BCL-2), development of rational combination regimens, expansion into earlier treatment settings, and long-term safety assessments of novel agents.

In summary, while targeted therapies remain largely investigational, tarlatamab and ADCs represent the most promising avenues for improving outcomes in SCLC. Sustained investment in biomarker research and innovative trial designs will be essential to achieving meaningful clinical advances.

## 7. Conclusions

SCLC remains a highly aggressive malignancy with limited therapeutic progress over recent decades. While conventional treatment is still dominated by chemoimmunotherapy, novel agents such as tarlatamab and ADCs are beginning to redefine second-line options, offering improved outcomes and manageable toxicity. However, challenges persist, including rapid disease progression, limited biomarker integration, and toxicity concerns with targeted therapies. Future advances will depend on biomarker-driven strategies, refined combination regimens, and rational trial design. With ongoing research, particularly in molecular profiling and immunotherapeutic innovation, the outlook for patients with SCLC may improve meaningfully in the years ahead.

## Figures and Tables

**Figure 1 cancers-17-02256-f001:**
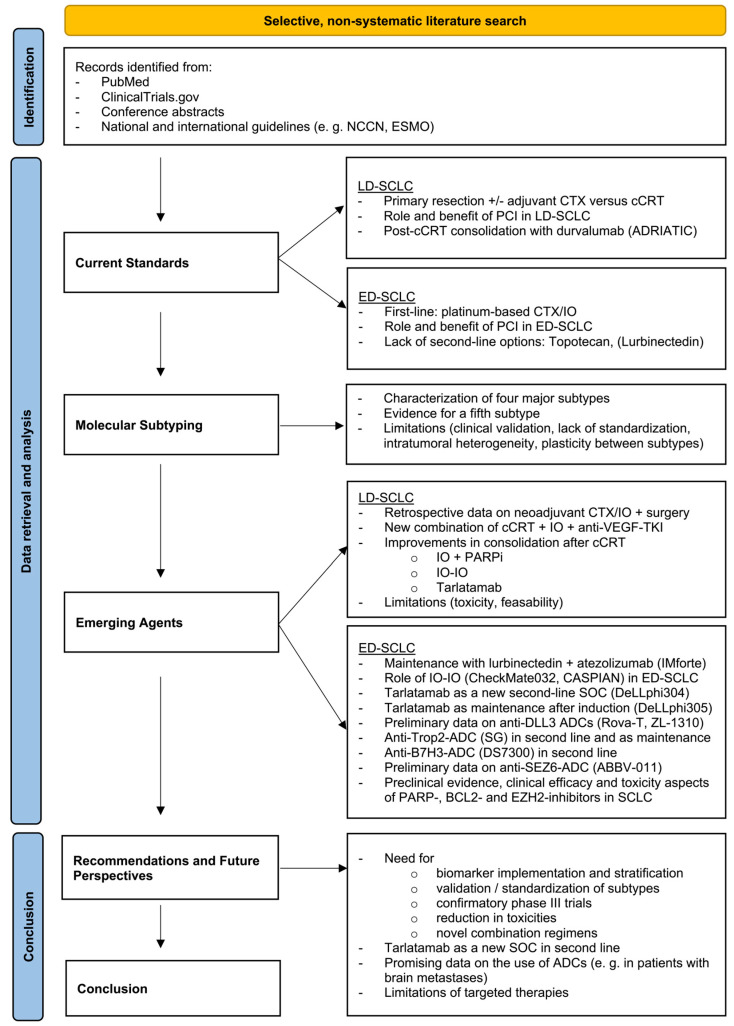
Flowchart illustrating the thematic structure of this review. Abbreviations used in this figure include ADCs (antibody-drug conjugates), BCL-2 (B-cell lymphoma 2), B7-H3 (B7 Homolog 3, CD276), cCRT (concurrent chemoradiotherapy), CTX (chemotherapy), DLL3 (Delta-like protein 3), ED (Extensive Disease), EZH2 (enhancer of zeste homolog 2), IO (immunotherapy), LD (Limited Disease), PARPi (poly ADP-ribose polymerase inhibitor), SEZ6 (seizure protein 6 homolog), SG (sacituzumab govitecan), SOC (standard of care), TKI (Tyrosine Kinase Inhibitor), Trop-2 (Tumor-associated calcium signal transducer 2), and VEGF (Vascular Endothelial Growth Factor).

**Figure 2 cancers-17-02256-f002:**
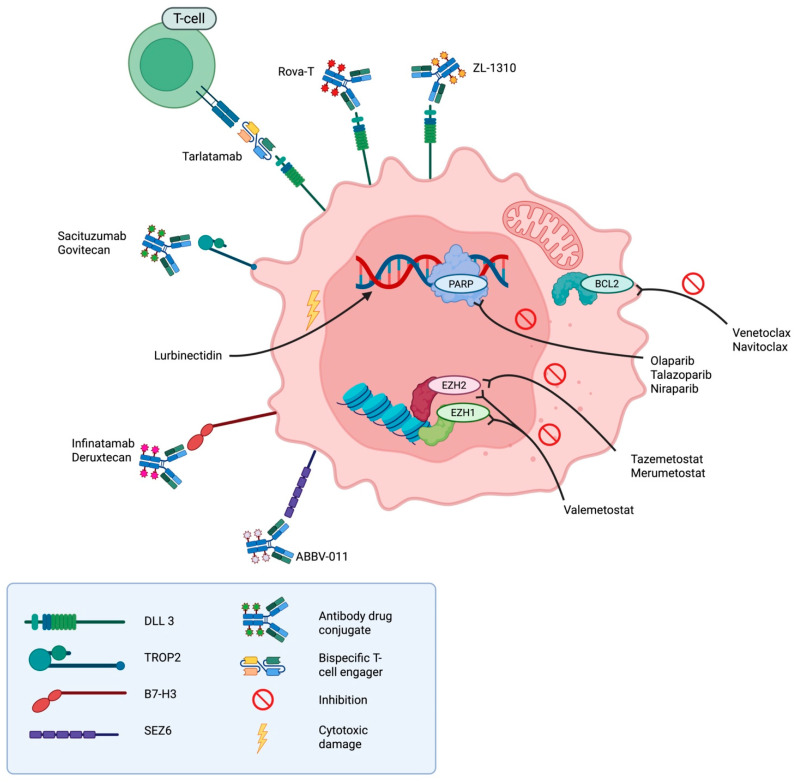
Graphical overview of emerging therapeutic strategies in small cell lung cancer (SCLC). This schematic illustrates key classes of novel agents under investigation for the treatment of SCLC. Antibody-drug conjugates (ADCs) such as Rova-T, ZL-1310, sacituzumab govitecan, infinatamab-deruxtecan, and ABBV-011 deliver targeted cytotoxic payloads to tumor-associated antigens. Tarlatamab represents a novel bispecific T-cell engager targeting DLL3. Additional compounds include inhibitors of anti-apoptotic BCL2 proteins (venetoclax, navitoclax), epigenetic regulators EZH1/2 (tazemetostat, merumetostat, valemetostat), and DNA repair enzymes PARP (olaparib, talazoparib, niraparib). Lurbinectedin, a cytotoxic agent, induces DNA damage independently of targeted pathways. Image created in BioRender. Reitnauer, L. (2025) https://BioRender.com/miyx8lj (accessed on 25 May 2025).

**Table 1 cancers-17-02256-t001:** Overview of the TNM classification and UICC 8th edition staging system for small cell lung cancer (SCLC), including 5-year survival rates based on data from the National Cancer Database (NCDB) [4].

TNM/UICC Stage	Tumor Characteristics	5-Year Survival Rate
**Limited Disease**		
Stage I (IA, IB)	Early-stage tumor: T1–T2 (≤5 cm or >5 cm but confined to one lobe), N0, M0	~13–25%
Stage II (IIA, IIB)	Locally advanced tumor: T2–T3 (infiltrating adjacent structures), N0 or N1, M0	~17–21%
Stage III (IIIA, IIIB, IIIC)	Advanced local tumor: T3–T4 (e.g., invasion of chest wall, vessels, or other lobes), N1–N3, M0	~9–13%
**Extensive Disease**		
Stage IV	Presence of distant metastases (M1)	<5%

**Table 2 cancers-17-02256-t002:** Overview of clinical trials and recent approvals in the treatment of small cell lung cancer (SCLC). Abbreviations used in this table include cCRT (concurrent chemoradiotherapy), CTFI (chemotherapy free interval), ED (Extensive Disease), LD (Limited Disease), OS (overall survival), PD1 (programmed death 1 protein) PD-L1 (programmed death ligand 1 protein), and SOC (standard of care).

Trial	N	Interventional Arm	Control Arm	Indication	Key Results	Limitations	Clinical Implications	Reference
IMpower133 (phase I/III)	403	Atezolizumab + carboplatin/ etoposide	Placebo + carboplatin/ etoposide	First-line, ED-SCLC	Significant improvement in OS: 12.3 vs. 10.3 months	Limited biomarker stratification	First-line anti-PD-L1 treatment option in ED-SCLC (SOC)	[37]
CASPIAN (phase III)	805	Durvalumab + platinum/etoposide (±tremelimumab)	Platinum/ etoposide	First-line, ED-SCLC	Significant improvement in OS: 13.0 vs. 10.5 months	Addition of tremelimumab without extra benefit	First-line anti-PD-L1 treatment option in ED-SCLC (SOC)	[38]
ASTRUM-005 (phase III)	585	Serplulimab + carboplatin/ etoposide	Placebo + carboplatin/ etoposide	First-line, ED-SCLC	Significant improvement in OS: 15.4 vs. 10.9 months	No direct comparison to PD-L1 inhibitors	Novel anti-PD1 first-line treatment option in ED-SCLC	[39]
ADRIATIC (phase III)	600	Durvalumab (±tremelimumab)	Placebo	Consolidation post-cCRT, LD-SCLC	Significant improvement in OS: 55.9 vs. 33.4 months	Data on dual checkpoint inhibition still pending	SOC for consolidation therapy post-cCRT in LD-SCLC	[36]
ATLANTIS (phase III)	613	Lurbinectedin + doxorubicin	Topotecan or CAV (cyclophosphamide, doxorubicin, vincristine)	Second-line SCLC	No significant improvement in OS: 8.6 vs. 7.6 months	No survival benefit; heterogeneity in patient selection; no approval	Signs of better tolerability; CTFI proved as prognostic/predictive	[40,41]

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
