# Peer review of "Redefining the Fight Against SCLC: Standards, Innovations, and New Horizons"

_cancers, 2025, doi:10.3390/cancers17132256_

Round 1
Reviewer 1 Report
Comments and Suggestions for Authors
This article addresses a critical topic in thoracic oncology, providing a comprehensive overview of small cell lung cancer (SCLC) epidemiology, molecular pathology, diagnosis, staging, treatments and emerging therapies. However, I have some concerns where the manuscript could be improved to enhace its clarity and coherence.
Major comments:
- Unnecessary methodology section: as a non-systematic narrative review, the inclusion of a "material an methods" section is inappropiate and may confuse readers expeting a systematic approach. I recommend removing this section and briefly stating in to the introduction that the review synthesizes current literature from pubmed, clinicaltrials.gov, and several guidelines without a formal protocol.
- Structural organization: There is not a clear progression between sections. I recommend restructuring the manuscript to group diagnosis and staging into a single section, followed by separate subsections for LD-SCLC and ED-SCLC treatments. The section for molecular subtyping could be before the treatment.
- Repetition: The introduction is too long and repetitive. Other several points, such as the the ADRIATIC trial results, are repeated across sections. It is necessary to consolidate the information across different sections.
- Imprecise terminology: The use of "first line" for LD-SCLC treatments is misleading, as this term is typically reserved for systemic therapy in advanced disease. I suggest using "primary treatment" for LD-SCLC. Additionally, the prognosis section conflates disease inherent prognostic factores (e.g. paraneoplastic syndrome) with treatment related outcomes (e.g. the use of PCI). These should be separated in distinct subsections. Perhaps, one section section could discuss the use of PCI and/or primary tumor radiotherapy in localized and extensive disease.
Author Response
Comment 1: Unnecessary methodology section: as a non-systematic narrative review, the inclusion of a "material an methods" section is inappropiate and may confuse readers expeting a systematic approach. I recommend removing this section and briefly stating in to the introduction that the review synthesizes current literature from pubmed, clinicaltrials.gov, and several guidelines without a formal protocol.
Response 1: We appreciate the reviewer’s comment and understand the concern regarding the potential confusion caused by including a "Materials and Methods" section in a narrative review. However, following internal discussion among the co-authors, and in response to conflicting comments between the reviewers and a request for greater transparency regarding the literature selection process, we have opted to retain a concise methodology section. We have revised its wording to clearly emphasize the non-systematic nature of the review and to avoid any implication of a formal systematic approach. This section now explicitly states that the review is based on a selective literature search of PubMed, ClinicalTrials.gov, and major guideline documents, without the use of a predefined protocol or quantitative synthesis. We hope this strikes an appropriate balance between transparency and clarity for the reader.
Comment 2: Structural organization: There is not a clear progression between sections. I recommend restructuring the manuscript to group diagnosis and staging into a single section, followed by separate subsections for LD-SCLC and ED-SCLC treatments. The section for molecular subtyping could be before the treatment.
Response 2: We appreciate the reviewer’s helpful suggestion regarding the manuscript’s structure. In response, we have revised the organization to improve clarity and logical flow. Specifically, we have now combined the diagnostic and staging information into a single dedicated section early in the manuscript. The treatment sections have been restructured into two distinct subsections focusing on limited-stage (LD-SCLC) and extensive-stage (ED-SCLC) disease, respectively. Furthermore, the section on molecular subtyping has been moved before the treatment discussion to better reflect its emerging relevance for therapeutic stratification. We believe these changes enhance the manuscript’s readability and coherence and thank the reviewer for this valuable recommendation.
Comment 3: Repetition: The introduction is too long and repetitive. Other several points, such as the the ADRIATIC trial results, are repeated across sections. It is necessary to consolidate the information across different sections.
Response 3: We thank the reviewer for this valuable feedback. In the revised manuscript, we have shortened and streamlined the introduction to remove redundancies and improve focus. Additionally, we have carefully reviewed the full text to identify and eliminate repetitive content, particularly with regard to the ADRIATIC trial, which is now discussed in a single, consolidated section to avoid duplication. These changes have been made to enhance clarity, reduce overlap, and improve the overall structure and readability of the manuscript. However, in order to address comments from all reviewers, we also had to expand some parts of the introduction such as the epidemiology section.
Comment 4: Imprecise terminology: The use of "first line" for LD-SCLC treatments is misleading, as this term is typically reserved for systemic therapy in advanced disease. I suggest using "primary treatment" for LD-SCLC. Additionally, the prognosis section conflates disease inherent prognostic factores (e.g. paraneoplastic syndrome) with treatment related outcomes (e.g. the use of PCI). These should be separated in distinct subsections. Perhaps, one section section could discuss the use of PCI and/or primary tumor radiotherapy in localized and extensive disease.
Response 4: We thank the reviewer for this valuable comment. In response, we have removed the term "first-line treatment" when referring to therapeutic approaches for limited-stage SCLC to align with appropriate clinical terminology. Additionally, the prognosis section has been removed to avoid conflating disease-inherent and treatment-related factors. Relevant content on prophylactic cranial irradiation (PCI) has been relocated to the therapeutic sections for both limited-stage and extensive-stage disease, where it is discussed in the appropriate clinical context. We believe these adjustments improve both the accuracy and clarity of the manuscript.
Reviewer 2 Report
Comments and Suggestions for Authors
Nicely drafted review article submitted by the author. However, some specific area may add to justify the article title. My specific comments are:
1. Author should provide updated demographic data of NSCLC.
2. What are the trend of new patient in developed countries and in developing countries, as well as in poor countries.
3. Author didn't provide any insight on the use of generative AI in NSCLC prognosis.
4. Author may provide a data based on last five year about the death of patients of NSCLC (First diagnosis, treatment start and death.
Author Response
Comment 1: Author should provide updated demographic data of NSCLC.
Response 1: We thank the reviewer for their suggestion. However, we believe there may have been a misunderstanding, as this review focuses on small cell lung cancer (SCLC) rather than non-small cell lung cancer (NSCLC). Accordingly, we have provided updated demographic data and incidence trends specifically for SCLC, as outlined in the revised manuscript. If the reviewer intended to refer to SCLC, we hope the new data address this point adequately.
Commont 2: What are the trend of new patient in developed countries and in developing countries, as well as in poor countries.
Response 2: We thank the reviewer for this insightful question. In the revised manuscript, we have addressed global trends in SCLC incidence across different regions. In high-income countries, such as the United States and those in Western Europe, the incidence of SCLC has been steadily declining over the past two decades, largely due to reduced smoking rates and improved tobacco control measures. In contrast, in many low- and middle-income countries (LMICs), the burden of SCLC remains substantial, and in some regions, incidence may be increasing due to continued high smoking prevalence, rising air pollution, and limited access to early detection and prevention programs. However, comprehensive data from low-resource settings remain limited and often there is no discrimination between lung cancer subtypes, underscoring the need for improved cancer surveillance and reporting infrastructure in these regions. We have added a summary of these global trends along with appropriate references.
Comment 3: Author didn't provide any insight on the use of generative AI in NSCLC prognosis.
Response 3: We thank the reviewer for this thoughtful comment. We have not included generative AI in this review, as its application in small cell lung cancer (SCLC) prognosis is not yet established in clinical practice and current data remain scarce. While generative AI is increasingly explored in oncology its role in SCLC remains largely investigational and warrants further research.
Comment 4: Author may provide a data based on last five year about the death of patients of NSCLC (First diagnosis, treatment start and death.
Response 4: We thank the reviewer for this suggestion. While detailed longitudinal data linking first diagnosis, treatment initiation, and time of death over the past five years are limited in published sources, we have addressed this point by including an updated sentence on 5-year survival rates for SCLC based on the most recent population-level data. This provides a relevant overview of long-term outcomes and reflects current trends in survival. Stage-dependent 5-year survival rates can also be found in Table 1 of the manuscript.
Reviewer 3 Report
Comments and Suggestions for Authors
The article provides a comprehensive review of small cell lung cancer (SCLC), discussing treatment modalities, recent innovations, and research directions. It highlights the aggressive nature of SCLC, molecular characteristics, diagnostic challenges, and evolving strategies like immunotherapy, targeted therapies, and liquid biopsy techniques. However, the manuscript lacks depth, and systematic methodology. Below are the major concerns for the major revision:
- Page 1—The abstract lacks a clear statement of the authors' main findings or unique contributions, requiring a concise statement of what they have investigated to add value. Moreover, try to maintain the abbreviation in the abstract using full words in the first addition and then use the abbreviated word.
- By the end of the introduction, add a flowchart showing all the steps and points that you have covered in your review.
- The methodology for article selection need to be explain in details. How were studies selected for inclusion, especially among conflicting results?
- The structure of the paper is repetitive and lacks logical flow. Sections such as “Prognosis and Biomarkers” and “Innovations and New Horizons” overlap significantly.
- Why was there no discussion of the poor concordance between tissue and ctDNA profiles in SCLC?
- The review presents multiple novel therapies (e.g., tarlatamab, ADCs, PARP inhibitors) with promising early-phase results but rarely discusses their limitations, toxicity profiles, or practical barriers to implementation in real-world settings. How do the authors reconcile the enthusiasm for these new agents with the high rates of adverse events, limited patient eligibility, or lack of biomarker-driven selection?
- Why did the authors poorly address the challenges of interconversion between subtypes?
- How do the authors reconcile the fact that most subtype definitions are still under development and lack standardized assays?
- What evidence supports the claim that SCLC-I and SCLC-Y respond better to chemoimmunotherapy?
- How do the authors justify promoting agents like ZL-1310, which are still in phase I with only preliminary results?
- Why were early-phase studies given equal weight to mature phase III data?
- The similarity of the paper is too high (31%), please work in this.
- Figure 1 needs enhancements in their quality and should be over 300 DPI.
- Add a new section as recommendations and future works which explain the main current problems, limitations, and others and what the authors recommend for these specific points.
- Pages 15, in the conclusion, as it is a review work so you need to provide main findings in a way that explain your finding for the review not add a general finding or what you did. It does not reflect on the implications of the findings for policymakers, industry stakeholders, or researchers working in the field.
- Add a table of nomenclature defining all the abbreviations used in the paper.
- Double check for the grammar.
Author Response
Comment 1: Page 1—The abstract lacks a clear statement of the authors' main findings or unique contributions, requiring a concise statement of what they have investigated to add value. Moreover, try to maintain the abbreviation in the abstract using full words in the first addition and then use the abbreviated word.
Response 1: We thank the reviewer for this helpful feedback. In the revised abstract, we have added a clearer statement highlighting the main findings and contributions of our review, the emerging role of novel agents such as tarlatamab and the potential clinical relevance of molecular subtyping in SCLC. Additionally, we have revised the use of abbreviations to ensure that all terms are written in full upon first mention, followed by their abbreviated forms. We also included a table of nomenclature in the supplementary material section. We believe these changes enhance the clarity and value of the abstract and appreciate the reviewer’s guidance.
Comment 2: By the end of the introduction, add a flowchart showing all the steps and points that you have covered in your review.
Response 2: We thank the reviewer for this helpful suggestion. In response, we have added a flowchart at the end of the introduction that outlines the structure of the review. The flowchart summarizes the main topics covered, including diagnosis and staging, molecular subtyping, treatment of limited-stage and extensive-stage SCLC, and emerging novel therapies. We believe this visual overview improves the clarity and navigability of the manuscript.
Comment 3: The methodology for article selection need to be explain in details. How were studies selected for inclusion, especially among conflicting results?
Response 3: We thank the reviewer for this important observation. Unfortunately there have been conflicting comments between reviewers regarding the methodology section. Therefore, in the revised manuscript, we have adjusted the methodology section according to all reviewer comments. As this is a narrative, non-systematic review, we did not follow a predefined protocol. Instead, we performed a selective literature search using PubMed and ClinicalTrials.gov, focusing on studies published between 2015 and 2025. Priority was given to peer-reviewed original research articles, high-impact reviews, and clinical trial reports that were relevant to the current therapeutic landscape. In cases of conflicting evidence, we critically appraised study design, sample size, and recency, and we aimed to reflect divergent findings in a balanced manner. We have clarified this methodology in the revised version to enhance transparency for the reader.
Comment 4: The structure of the paper is repetitive and lacks logical flow. Sections such as “Prognosis and fers” and “Innovations and New Horizons” overlap significantly.
Response 4: We appreciate the reviewer’s observation regarding structural redundancy and lack of logical flow. In response, we have revised the manuscript to streamline and reorganize overlapping content. Specifically, the former sections “Prognosis and Therapeutic Frontiers” and “Innovations and New Horizons” have been consolidated and restructured to eliminate repetition and improve coherence. The revised structure now follows a more logical progression from diagnosis and staging, to molecular subtyping, current treatment strategies, and finally recommendations and future perspectives. These changes aim to enhance clarity, avoid duplication, and provide a more concise and focused narrative.
Comment 5: Why was there no discussion of the poor concordance between tissue and ctDNA profiles in SCLC?
Response 5: We thank the reviewer for raising this important point. In the previous version of the manuscript, we included a section discussing the potential role of ctDNA in SCLC. However, after internal review, we decided to remove this section due to the currently limited and heterogeneous evidence base, as well as the lack of consistent clinical applicability in SCLC at this time. We agree that poor concordance between ctDNA and tissue is a relevant challenge, but data for SCLC is rather limited compared to NSCLC. Therefore we decided to remove this section.
Comment 6: The review presents multiple novel therapies (e.g., tarlatamab, ADCs, PARP inhibitors) with promising early-phase results but rarely discusses their limitations, toxicity profiles, or practical barriers to implementation in real-world settings. How do the authors reconcile the enthusiasm for these new agents with the high rates of adverse events, limited patient eligibility, or lack of biomarker-driven selection?
Response 6: Thank you for this important comment. We agree that while novel therapies such as tarlatamab, ADCs, and PARP inhibitors have shown encouraging early-phase results, their clinical implementation faces significant challenges. In response, we have expanded the manuscript to critically address these issues. Specifically, we now discuss the high incidence of treatment-related adverse events observed with dual checkpoint inhibition and certain targeted therapies, which may limit their use to select patient populations with good performance status. Additionally, we acknowledge that many of these agents lack validated predictive biomarkers, complicating patient selection and limiting their applicability in routine clinical practice. We have also added a section on current limitations and future directions, highlighting the need for biomarker-driven trials, long-term safety data, and real-world validation to ensure that these promising approaches can be translated into meaningful clinical benefit.
Comment 7: Why did the authors poorly address the challenges of interconversion between subtypes?
Response 7: We appreciate this important comment. In the revised manuscript, we now more clearly acknowledge the dynamic plasticity of SCLC subtypes and the potential for interconversion, particularly under therapeutic pressure. While emerging preclinical models and longitudinal biopsy studies suggest that SCLC subtypes can shift in response to treatment, robust clinical data remain limited. We have added a section discussing the biological and methodological challenges in capturing these transitions, including tumor heterogeneity, sampling limitations, and a lack of standardized longitudinal assessment tools.
Comment 8: How do the authors reconcile the fact that most subtype definitions are still under development and lack standardized assays?
Response 8: We agree that current SCLC subtype definitions are largely based on transcriptomic profiles derived from preclinical models and retrospective analyses, and lack harmonized, clinically validated assays. In the revised version, we have expanded our discussion to emphasize the experimental nature of these classifications. We also highlight ongoing efforts to translate molecular subtypes into clinically actionable diagnostics, while cautioning that prospective validation is still needed before routine implementation.
Comment 9: What evidence supports the claim that SCLC-I and SCLC-Y respond better to chemoimmunotherapy?
Response 9: Emerging evidence from retrospective analyses and translational studies supports this claim. For example, retrospective data from the IMpower133 trial suggest that SCLC-I tumors, characterized by an inflamed transcriptomic profile and increased immune cell infiltration, are more responsive to immunotherapy. Similarly, the SCLC-Y subtype has shown enrichment for YAP1 expression, which has been linked to sensitivity to DNA-damaging agents and immunotherapy in some preclinical models. We have updated the manuscript to include these references and clarify that these findings, while promising, require further validation in prospective trials.
Comment 10: How do the authors justify promoting agents like ZL-1310, which are still in phase I with only preliminary results?
Response 10: Thank you for this comment. We acknowledge that ZL-1310 is currently in early-phase clinical development, and our intention is not to promote it prematurely but rather to provide a comprehensive overview of emerging therapeutic SCLC. We have revised the manuscript to clarify that the data for ZL-1310 are preliminary and based on a small phase I cohort. While the initial response rates are promising, we emphasize that these findings require validation in larger, randomized trials before any clinical recommendations can be made. Our revised text now reflects a more cautious interpretation and clearly distinguishes between established therapies and agents that remain investigational.
Comment 11: Why were early-phase studies given equal weight to mature phase III data?
Response 11: Thank you for your important observation. We agree that phase III data carry greater weight in guiding clinical practice. In the revised manuscript, we have adjusted the structure and wording to better differentiate between mature evidence from randomized phase III trials (e.g., DeLLphi-304, IMforte) and preliminary findings from early-phase studies. Our aim was to provide a comprehensive overview of the evolving therapeutic landscape in SCLC, which is still largely shaped by early-stage research due to the historically limited treatment options. However, we now explicitly indicate the developmental stage of each agent discussed and clarify that enthusiasm for early-phase results should be tempered by the need for confirmatory data before clinical integration.
Comment 12: The similarity of the paper is too high (31%), please work in this.
Response 12: We thank the reviewer for pointing this out. In response, we have thoroughly revised the manuscript to reduce textual similarity and ensure originality. Overlapping or closely paraphrased content has been reworded, condensed, or replaced with original synthesis. We have also reviewed citation practices to ensure appropriate attribution. Following these revisions, we expect a significant reduction in overall similarity while maintaining scientific accuracy and clarity.
Comment 13: Figure 1 needs enhancements in their quality and should be over 300 DPI.
Response 13: We thank the reviewer for this important feedback. Figure 1 has been updated and enhanced to ensure improved visual clarity and resolution. The revised figure now meets publication standards with a resolution exceeding 300 DPI. We have also reviewed the formatting to ensure consistency with journal guidelines and improved overall readability.
Comment 14: Add a new section as recommendations and future works which explain the main current problems, limitations, and others and what the authors recommend for these specific points.
Response 14: We appreciate the reviewer’s suggestion to strengthen the manuscript by adding a dedicated section on recommendations and future directions. In response, we have included a new section titled "Recommendations and Future Perspectives". This section outlines the current challenges in the field, such as treatment resistance, limited biomarkers, and the need for standardized molecular subtyping, and provides concrete recommendations for future research, including the validation of emerging therapies in prospective trials and the development of subtype-specific treatment strategies. We believe this addition adds important context and forward-looking value to the review.
Comment 15: Pages 15, in the conclusion, as it is a review work so you need to provide main findings in a way that explain your finding for the review not add a general finding or what you did. It does not reflect on the implications of the findings for policymakers, industry stakeholders, or researchers working in the field.
Response 15: We thank the reviewer for this constructive feedback. In the revised conclusion (page 15), we have restructured the text to clearly summarize the main findings of the review, highlighting specific advances in therapeutic strategies, the emerging relevance of molecular subtyping, and the potential of novel agents such as tarlatamab and ADCs in SCLC. Additionally, we have concentrated the conclusion in order to emphasize the need for biomarker-driven clinical trial designs, investment in translational research, and the development of standardized diagnostic tools to support precision medicine approaches in SCLC.
Comment 16: Add a table of nomenclature defining all the abbreviations used in the paper.
Response 16: We thank the reviewer for this helpful suggestion. In response, we have added a table of nomenclature at the end of the manuscript, defining all abbreviations and acronyms used throughout the text. This addition is intended to enhance readability and provide clarity for readers who may be less familiar with the terminology.
Comment 17: Double check for the grammar.
Response 17: We appreciate the reviewer’s attention to detail. The entire manuscript has been carefully proofread by different authors, and grammar, syntax, and punctuation have been thoroughly reviewed and corrected to ensure clarity, consistency, and adherence to academic writing standards.
Round 2
Reviewer 3 Report
Comments and Suggestions for Authors
The authors have improved their work according to my previous comments. However, several points still require their attention:
correct only the mistakes:
- The flowchart was developed in a confusing manner and does not fully illustrate the main topics covered in this work. I encourage the authors to review several published works that include flowcharts to understand how to scientifically and clearly present such visual summaries.
- In section or subsection headings, tables should not be referenced. For instance, Subsection 4.2 requires revision—please remove the mention of "Table 3" from the heading.
-Please provide more detailed explanations regarding the works summarized in Tables 1 and 2 . The authors should describe and analyze what each cited work contributes and highlight their main limitations. If possible, the limitations of the works included in Table 2 can be added directly within the table for better clarity and comparison.
- Consider converting the recommendations and future work into bullet points or numbered items to make them more specific and easier to follow.
-Review and revise all references to ensure they fully conform to the journal’s citation style guidelines. This includes checking formatting for author names, journal titles, years, volume/issue numbers, DOIs, and consistency across all entries.
Author Response
The authors have improved their work according to my previous comments. However, several points still require their attention:
correct only the mistakes:
Comment 1: The flowchart was developed in a confusing manner and does not fully illustrate the main topics covered in this work. I encourage the authors to review several published works that include flowcharts to understand how to scientifically and clearly present such visual summaries.
Response 1: We thank the reviewer for this feedback. We have revised the flowchart (Figure 1) to more clearly reflect the thematic structure of the manuscript. The updated version now illustrates the methodological approach of this review and incorporates current therapeutic standards, molecular subtyping, as well as emerging therapeutic strategies such as bispecific antibodies, ADCs, and targeted agents. In doing so, we aimed to improve both scientific clarity and visual logic. While we carefully reviewed published examples of narrative review figures, we were uncertain about the reviewer’s specific expectations. We therefore focused on enhancing conceptual coherence and readability while maintaining alignment with the structure of the manuscript. We hope that the revised flowchart now meets the reviewer’s expectations.
Comment 2: In section or subsection headings, tables should not be referenced. For instance, Subsection 4.2 requires revision—please remove the mention of "Table 3" from the heading.
Response 2: We thank the reviewer for this important observation. We have revised the heading of Subsection 4.2 by removing the reference to "Table 3" to ensure consistency with formatting guidelines. The table is now appropriately referenced within the body text of the section.
Comment 3: Please provide more detailed explanations regarding the works summarized in Tables 1 and 2 . The authors should describe and analyze what each cited work contributes and highlight their main limitations. If possible, the limitations of the works included in Table 2 can be added directly within the table for better clarity and comparison.
Response 3: Thank you for this valuable feedback. We believe you are referring to Tables 2 and 3 rather than Table 1. Accordingly, we have revised the manuscript to provide more detailed explanations of the studies summarized in Tables 2 and 3, describing the specific clinical implications and main findings of each work. We have also highlighted the key limitations of each study to offer a more balanced critical analysis. Additionally, we have added dedicated columns in Tables 2 and 3 summarizing these limitations and contributions to enhance clarity and facilitate direct comparison. We believe these changes significantly improve the rigor and readability of the discussion.
Comment 4: Consider converting the recommendations and future work into bullet points or numbered items to make them more specific and easier to follow.
Response 4: Thank you for this helpful suggestion. We agree that presenting the recommendations in bullet-point form improves clarity and readability. We have revised the relevant section to include clearly delineated bullet points summarizing the key recommendations and priorities for future research. We believe this change makes the text easier to follow and more accessible to readers.
Comment 5: Review and revise all references to ensure they fully conform to the journal’s citation style guidelines. This includes checking formatting for author names, journal titles, years, volume/issue numbers, DOIs, and consistency across all entries.
Response 5: Thank you for pointing this out. We have carefully reviewed and revised all references to ensure full conformity with the journal’s citation style guidelines. This includes standardizing author name formats, journal titles, publication years, volume and issue numbers, and DOIs. We have also checked for consistency across all entries to improve overall accuracy and presentation.